# Tumor Treating Fields Alter the Kinomic Landscape in Glioblastoma Revealing Therapeutic Vulnerabilities

**DOI:** 10.3390/cells12172171

**Published:** 2023-08-30

**Authors:** Amber B. Jones, Taylor L. Schanel, Mikayla R. Rigsby, Corinne E. Griguer, Braden C. McFarland, Joshua C. Anderson, Christopher D. Willey, Anita B. Hjelmeland

**Affiliations:** 1Department of Cell, Developmental and Integrative Biology, Heersink School of Medicine, University of Alabama at Birmingham, Birmingham, AL 35233, USA; amberj96@uab.edu (A.B.J.); mikaylar@uab.edu (M.R.R.); bdcox@uab.edu (B.C.M.); 2Department of Radiation Oncology, Heersink School of Medicine, University of Alabama at Birmingham, Birmingham, AL 35233, USA; tschanel@uab.edu (T.L.S.); janders7@uab.edu (J.C.A.); 3Department of Radiation Oncology, University of Iowa, Iowa City, IA 52242, USA; cgriguer@healthcare.uiowa.edu

**Keywords:** glioblastoma, tumor treating fields, temozolomide resistance, kinomics

## Abstract

Treatment for the deadly brain tumor glioblastoma (GBM) has been improved through the non-invasive addition of alternating electric fields, called tumor treating fields (TTFields). Improving both progression-free and overall survival, TTFields are currently approved for treatment of recurrent GBMs as a monotherapy and in the adjuvant setting alongside TMZ for newly diagnosed GBMs. These TTFields are known to inhibit mitosis, but the full molecular impact of TTFields remains undetermined. Therefore, we sought to understand the ability of TTFields to disrupt the growth patterns of and induce kinomic landscape shifts in TMZ-sensitive and -resistant GBM cells. We determined that TTFields significantly decreased the growth of TMZ-sensitive and -resistant cells. Kinomic profiling predicted kinases that were induced or repressed by TTFields, suggesting possible therapy-specific vulnerabilities. Serving as a potential pro-survival mechanism for TTFields, kinomics predicted the increased activity of platelet-derived growth-factor receptor alpha (PDGFRα). We demonstrated that the addition of the PDGFR inhibitor, crenolanib, to TTFields further reduced cell growth in comparison to either treatment alone. Collectively, our data suggest the efficacy of TTFields in vitro and identify common signaling responses to TTFields in TMZ-sensitive and -resistant populations, which may support more personalized medicine approaches.

## 1. Introduction

With a five-year survival rate of less than five percent, glioblastoma (GBM) remains one of the most difficult malignancies to treat [1]. Despite decades of research, there have been minimal advances in the standard-of-care treatment of GBM, consisting of surgical resection and adjuvant radio- and chemotherapy with the DNA alkylating agent, temozolomide (TMZ) [2]. Following this aggressive initial treatment regimen, GBMs often recur with an outgrowth of therapy-resistant cell populations, and improved treatments for primary and recurrent GBMs are needed [3,4,5,6]. A newly implemented treatment that may aid in combating GBM-tumor progression and therapeutic resistance is the delivery of alternating electrical fields, more commonly known as tumor treating fields (TTFields) [7,8].

These TTFields were first approved in 2011, for recurrent GBM, and then in 2015, for newly diagnosed GBM [9,10]. Since then, TTFields have shown extraordinary clinical promise in extending overall patient survival by approximately 5 months when used with maintenance TMZ with no reported off-target toxicities to normal surrounding brain tissue [7]. A recent meta-analysis investigating clinical outcomes utilizing TTFields reported that, based on 1309 cases spanning 14 studies, there was a significant increase in one-year survival rates for TTFields-treated patients (>60%) compared to untreated patients (<40%), warranting their continued utilization [11]. Guzauskas et al. further provided an integrated epidemiological approach using TTFields EF-14 clinical-trial data to predict the survival probability of GBM patients [12]. Based on their analysis, it was predicted that patients alive two years after starting TTFields have a 20.7% probability of surviving for 10 years after diagnosis [12]. Overall, there is evidence to support further investigation into and the continued implementation of TTFields for clinical use with GBM patients.

Using a clinically relevant frequency range of 100 kHz to 500 kHz and an intensity of 1–3 V/cm, TTFields were shown to preferentially perturb mitotic progression in actively dividing cells [13,14]. More specifically, by forcing their alignment towards the direction of the electrical fields, TTFields negatively affect tubulin polymerization, disrupting proper mitotic spindle formation in cancer cells [15,16]. Additionally, TTFields disturb the localization of septin, which is a GTP-binding protein crucial for accurate cleavage-furrow organization, contributing to irregular cell division and mitotic catastrophe [13,17]. Further contributing to this effect, due the polarity of the macromolecules and organelles associated with mitosis, TTFields forcibly cause the aggregation of these structures at the cleavage furrow, preventing the proper cytokinesis of daughter cells [13]. Collectively, there is strong evidence that TTFields serve as anti-mitotic agents, and there are increasing studies highlighting their implications in other cancer-relevant pathways [18]. However, there have been limited studies addressing whether and how TTFields disproportionally affect therapy-resistant GBM cells. Our studies sought to identify differential growth patterns in TMZ-resistant GBM cells and target the potential signaling cascades in response to these effects.

Kinases serve as signaling mediators that are often targeted in patients with FDA-approved drugs or during clinical trials with novel inhibitors. An understanding of the kinases altered by TTFields could enable the design of novel combinatorial therapies or identify biomarkers for response. We therefore performed kinomic profiling using TMZ-sensitive and -resistant GBM models to determine potential signaling cascades that may be influenced by TTFields.

## 2. Materials and Methods

### 2.1. Cell Models and Culture

Primary and TMZ-resistant GBM-patient-derived xenografts (JX22P and JX22T, respectively) were generously provided by Dr. Jann Sarkaria (Mayo Clinic; https://www.mayo.edu/research/labs/translational-neuro-oncology/mayo-clinic-brain-tumor-patient-derived-xenograft-national-resource/pdx-characteristics/pdx-phenotype, accessed on 5 July 2023). Parental and TMZ-resistant U251 cells (U251P and U251T, respectively) were generated by Dr. Corrine Griguer, as previously described [19]. To preserve molecular similarities to the primary human tumor, Jx22 PDX cells were passaged in Balbc nu/nu mice as subcutaneous engraftments, in accordance with approval from the UAB Institutional Animal Care and Use Committee (IACUC). Tumors were harvested and dissociated using the Worthington Papain dissociation system, as previously described [20,21]. Cells isolated from PDX and U251 cells were both cultured under serum-free conditions as neurospheres, as we described previously. The DMEM/F12 medium (Life Technologies, cat. no. 21041-025) was used as a base, supplemented with Gem21 Neuroplex without Vitamin A (Gemini Bioproducts, Sacramento, CA, USA, cat. no. 400-161), 100 U/mL penicillin, and 100µg/mL of streptomycin (Gibco, Billings, MT, USA, cat. no. 15-140-122), 1% sodium pyruvate (Gibco, cat. no. 11360070), and 10 ng/mL of both epidermal growth factor and fibroblast growth factor (Gemini Bioproducts, cat. nos. 300-110P and 300-112P).

### 2.2. TTFields-Based Growth Analysis

To perform growth assessments in the presence of TTFields, we utilized the inovitro^TM^ Laboratory Research System graciously provided by Novocure. Cells were plated on 22 mm plastic coverslips (Thermo Fisher, Waltham, MA, USA, cat. no. 175977) coated with geltrex at a density of 5 × 10^4^ in a 200 µL drop suspension of cell-culture medium. Following overnight adhesion and recovery, coverslips containing cells were then placed in ceramic dishes that contained conductive arrays supplied with 2 mL of fresh media and sealed with parafilm. The ceramic dishes were then moved to the provided base plate, which was connected to the TTFields power supply. The base plate was subsequently placed into a cooling (18 °C, 5% CO_2_) incubator that aided in regulating the emitted heat generated from the TTFields. The system was then set to administer TTFields at a target frequency of 200 kHz, target intensity of 1 V/cm, and target dish temperature of 37 °C, which was monitored using the associated Novocure inovitro computer software (last modified 6 June 2019, Novocure, Switzerland) [22]. Furthermore, each experiment was performed with a matched pair of treatment groups that were plated equally on coverslips but maintained in 6-well plates under standard incubation conditions (37 °C, 5% CO_2_) without exposure to TTFields. All experimental conditions (with or without TTFields applied) had daily media changes. Following completion of the experimental timeframe, coverslips were washed with PBS (Gibco, cat. no. 10010-023), formalin-fixed (Fisher, cat. no. 305-510), and then stained with a 1% crystal violet solution (Fisher Scientific, Waltham, MA, USA, cat. no. S25274B). Coverslips were first qualitatively imaged using an EVOS XL Core microscope (Life Technologies, Carlbad, CA, USA) and then quantitatively measured after solubilization with 10% acetic acid (Fisher Scientific, cat. no. A38S-500), and absorbance (562 nm) was read using a Synergy H1 Biotek microplate reader.

### 2.3. Kinomics

In collaboration with the UAB Kinome Core, kinomic analysis was performed as previously described [23,24,25]. In brief, protein lysates from control and TTFields-treated cells (total of 3 × 10^5^ cells per condition pooled from 3 replicate coverslips containing 1 × 10^5^ cells each) were harvested after 4 h using MPER lysis buffer (Thermo Fisher, cat. no. 78501) containing 1:100 Halt protease and phosphatase inhibitors (Invitrogen, Waltham, MA, USA, cat. no. 78440). After protein quantification with the BCA protein assay (Fisher Scientific, cat. no. 23227), lysates from the respective groups were loaded with kinase buffer onto the respective PamChip array (PTK (tyrosine kinome) or STK (serine/threonine kinome)). Protein lysates were then pumped through the respective porous arrays containing 196 phosphorylatable peptides or 144 serine and threonine phosphorylatable peptides. Detection of phosphorylation changes was determined using specific FITC conjugated antibodies where relative peptide phosphorylation signals were captured kinetically. Evolve (PamGene, Den Bosch, Netherlands, v3.1.0.5) and BioNavigator (Pamgene, v6.3.67.0) software were both utilized for analysis, and peptide alterations were provided for the respective treatment lysates. The significantly altered peptides were then uploaded to GeneGo Metacore (portal.genego.com, Clarivate accessed on 14 December 2022), at which point a network analysis was performed, followed by a cross comparison of these peptides’ predicted upstream kinases utilizing the UpKin software (PamGene Upstream Kinase Analysis-2018 v.6).

### 2.4. Lysate Preparation and Immunoblots

For immunoblotting, lysates were harvested like those used for kinomics, as detailed above. Lysed samples were passed through a 28-gauge insulin syringe and then spun down at 10,000 RPM for 10 min at 4 °C prior to quantifying protein levels with the BCA protein assay. After quantification, samples containing equivalent amounts of protein from control and TTFields-treated samples (40 µg) were prepared using Pierce Lane Marker 5× reducing sample buffer (Thermo Fisher, cat. no. 39000), with boiling at 95 °C for 5 min. Using Novex Wedge 4–20% Tris Glycine gels (Thermo Fisher, cat. no. XP04200BOXproduct number), gel electrophoresis was performed, and proteins were transferred to PVDF membranes (Bio-Rad, Hercules, CA, USA, cat. no. 1704272). Blocking was attained with 5% milk dissolved in Tris-buffered saline with 0.1% Tween^®^ 20 detergent (TBST) for 1 h at room temperature. Overnight incubation of primary antibodies at 4 °C was performed using the following antibodies: phosphoAKT (Ser129; Cell Signaling, Danver, MA, USA, cat. no. 13461), panAKT (Cell Signaling; cat. no. 4691), and GAPDH as a loading control (Cell Signaling, cat. no. 97166). Following overnight incubation of primary antibodies, IRDye 680CW goat anti-rabbit IgG or IRDye 800CW goat anti-mouse IgG secondary antibodies were incubated for 1 h at room temperature (Li-Cor 926-68071 and 926-32210). An Odyssey infrared imaging system (LI-COR) was utilized to image blots and Image Studio Lite v5.2.5 software was used to determine densitometry values for each respective protein sample.

### 2.5. Drug Treatments

To confirm TMZ sensitivity and resistance of GBM cell models, cells were seeded at 1 × 10^3^ cells per well in a 96-well plate. After overnight recovery, cells were treated with either DMSO vehicle control or 150 µM of TMZ (Sandoz, Basel, Switzerland, cat. no. 0781-2694-44). After 7 days, relative cell viability was assessed using the Cell Titer Glo 2.0 assay (Promega, Madison, WI, USA, cat. no. G9243) following manufacturer’s directions. For PDGFRA inhibition, initial IC50 values of crenolanib (Selleckchem, Houston, TX, USA, cat. no. CP-868596) were determined for cells independent of TTFields administration by treating cells with a serial dilution of drug for 72 h. After identification of approximate IC50s for the respective cell models, combinatorial treatment of crenolanib and TTFields was assessed by performing growth analysis as described above. Crenolanib was added to respective treatment groups concurrent with media changes.

2.6 Statistics: Experiments were performed in a minimum of biological triplicates, with replicate samples for each experimental condition. Data analysis was performed using Prism v9.5.0 (GraphPad) and the relevant statistical tests are indicated in the respective figure legends. Briefly, for basal growth analysis with individual treatment with TMZ or TTFields, a paired t-test was performed if the following assumptions were met: independence, continuous outcome variable, normal distribution (non-significant Shapiro–Wilk tests). For combined TTFields and crenolanib growth assessments, statistical analyses were performed using a one-way ANOVA with Tukey’s multiple comparison if the following assumptions were met: independence, normal distribution (non-significant Shapiro–Wilks tests), homogeneity of variance (non-significant Brown–Forsythe tests), and continuous outcome variable. Reported *p*-values were adjusted based on alpha = 0.05.

## 3. Results

### 3.1. Parental and TMZ-Resistant GBM Cells Have Reduced Growth in Response to TTField Treatment

To first assess the influence that TTFields have on GBM-cell growth in the context of TMZ sensitivity, we performed basal growth experiments using an in vitro TTFields system. Initially, we validated the TMZ resistance in our respective models following a 7-day treatment of TMZ, in which the Jx22T and U251T cell viability was not hindered by the treatment, while the parental cells displayed significantly reduced growth (Figure 1A,B). Next, we exposed the respective cell models to either controls or 200 kHz of TTFields for 72 h and determined the relative changes in cell growth based on crystal-violet staining (Figure 1C,D) and the corresponding quantification (Figure 1E,F). Our findings indicate a marked and significant overall reduction in cell growth in the groups treated with TTFields, which was independent of the TMZ sensitivity. However, we did notice that the extent of the cell-growth inhibition varied with the TMZ resistance. The percentage of cell growth in the Jx22T and U251T cells after the administration of the TTFields was greater than that of their parental counterparts (Figure 1E,F), suggesting some inherent differences in compensatory mechanisms in response to stress. Additionally, as the basal growth rates of the TMZ-resistant models were increased compared to their respective TMZ sensitivities, the TTFields sensitivity may have been independent of the cells’ proliferative capacities (Appendix A).

### 3.2. Kinomics Reveals Kinase Alterations Induced in TTFields-Treated Cells and Kinomics Reveals Kinase Alterations Induced in TTFields-Treated Cells

As both the parental and TMZ-resistant GBM cells had decreased growth in response to the TTFields, we next sought to determine whether similar kinase-signaling pathways were altered. Conveniently, kinomics leverages changes in phosphorylation events and computational modeling to identify signaling cascades activated or repressed by certain stimuli. Therefore, we utilized the PamStation12 platform to profile the kinome of parental and TMZ-resistant GBM cells with and without treatment with TTFields. We first identified notable differences in the overall patterns of the phosphorylation events upon TTFields treatment, which differed according to the cells’ TMZ sensitivity. For example, the treatment with TTFields decreased phosphorylation events in the TMZ-sensitive Jx22P cells in both the tyrosine (Figure 2A) and the serine/threonine (Figure 2B) phosphopeptide array; however, a reversal of this effect was observed in the TMZ-resistant Jx22T-TTFields-treated cells, with a general increase in phosphorylation events. Similar trends were also observed in the tyrosine array for the U251 model; however, the phosphorylation events were generally downregulated in the serine/threonine array in the TTFields-treated cells independently of the TMZ sensitivity in the U251 cells. This observation may reflect the differences due to the heterogeneity of the GBM or the differences between the patient-derived-xenograft and cell-line models.

To determine upstream kinase activity and kinase networks mediating TTFields-altered peptide phosphorylation, we utilized BioNaviator and GeneGo Metacore, respectively. Our analysis suggested a subset of kinases that were predicted to be activated (Figure 2C) or repressed (Figure 2D) by TTFields treatment. While there were kinases that were distinctly activated or repressed by TTFields in a cell-type-specific manner, there was a set of kinases that were commonly altered by TTFields across the four tested lines independently of TMZ sensitivity, including the TTFields-mediated activation of PDGFRα, TRKA/C, Tyro3/Sky LTK, Abl, Mer, and EphA2, and the TTFields repression of the activity of CK2α1, NuaK1, CK1 (α, δ, ε), ERK1, JNK1/2/3, PFTAIRE2, ERK2, and PKCγ/ε. We further identified that these TTFields-regulated kinases generated a signaling network that was centered around ERK activity, which can be exploited for potential global targeting (Figure 2E). Collectively, our kinomic analysis revealed TTFields-induced alterations in both TMZ-sensitive and -resistant GBM cells, and we identified kinase targets that can be mechanistically linked to TTFields responsiveness in cells.

### 3.3. CK2 Activity Decreased following Treatment with TTFields

Of the shared kinase targets that were predicted to be repressed by the TTFields treatment, CK2 was extensively linked to GBM lethality. This CK2 is a Ser/Thr kinase that has been shown to phosphorylate many different targets to regulate cell proliferation, migration, and angiogenesis and, further, to contribute to GBM aggressiveness [30,31]. One direct target of CK2 phosphorylation that has been used to monitor CK2 activity is the phosphorylation of AKT at serine 129 [32]. Using immunoblotting, we found that the TTFields-treated cells had decreased levels of serine 129 phosphorylated AKT but not total AKT, supporting the findings from our kinomic analysis (Appendix A). Together, the data indicate that TTFields can decrease CK2 activity.

### 3.4. Addition of PDGFR Inhibitor, Crenolanib, to TTFields Decreases GBM Growth Independently of Sensitivity to TMZ

Of the kinases predicted to be activated by TTFields, PDGFRA is a tyrosine-kinase receptor that is widely known to contribute to GBM growth and development. We hypothesized that the increase in PDGFRA activity may be a protective response to TTFields. To test this hypothesis, we used the blood–brain-barrier-penetrant PDGFRA/B inhibitor, crenolanib, due to the limited availability of selective PDGFRA inhibitors. We first determined the approximate half-maximal inhibitory (IC50) concentrations of crenolanib for the respective TMZ-sensitive and -resistant models (Appendix A). We then conducted a 72-h growth assessment of the control and TTFields-treated cell lines in the presence or absence of crenolanib (Figure 3). In both the TMZ-sensitive and the TMZ-resistant cells, there was a significant decrease in the growth of the GBM cells treated with both crenolanib and TTFields compared to either treatment alone.

We then further confirmed these findings using the more patient-representative Jx22 xenograft model displaying an even more pronounced reduction in cell growth when dually treated with both TTFields and crenolanib (Figure 4). These data confirm that PDGFR inhibition is a common vulnerability in GBM cells, independently of TMZ sensitivity. Overall, our study provides mechanistic insights into and the validation of pathways that can be exploited to further enhance the benefit of TTFields in treating GBM tumors and/or predicting therapeutic responses.

## 4. Discussion

The use of TTFields offers a unique opportunity to serve as a highly selective, non-invasive treatment option for GBM patients. As this treatment modality has been recognized for the clinical consideration of GBM tumors and its addition to the standard of care for GBM has been recommended, there is a need to further elucidate mechanisms that are regulated by TTFields to predict and improve its therapeutic efficacy. Here, we show that the growth of both TMZ-sensitive and TMZ-resistant GBM cells is significantly inhibited by TTField treatment. Commonly linked to TMZ resistance are the expression and activity of the DNA-repair enzyme, methylguanine methyltransferase (MGMT) [33,34]. Functionally, MGMT repairs TMZ-induced lesions on certain guanine and adenine residues, providing protection against TMZ-mediated cell death [35,36]. Although its potential role is uncertain, we hypothesize that the influence of TTFields on TMZ-resistant cell growth is most likely regulated by additional mechanisms, as GBM cells have a high degree of plasticity conferring their survival. In a 2017 study, Clark et al. displayed equal reductions in proliferation and neurosphere formation in both MGMT-expressing (TMZ-resistant) and MGMT-non-expressing (TMZ-sensitive) GBM cells in response to TTFields [37]. Additional studies by Silginer et al. and Fishman et al. suggest the efficacy of TTFields in TMZ-sensitive and -resistant populations [38,39]. Additionally, as TTFields were initially approved and successful against recurrent GBM in which TMZ-resistant (MGMT-expressing) cells are abundant, it is probable that the influence of MGMT status on TTFields sensitivity is negligible [7,8]. Therefore, we sought to identify alternative molecular mechanisms that may regulate GBM cells’ responsiveness to TTFields. We demonstrate that TTFields decrease CK2 activity and increase PDGFRA activity in a manner that is independent of sensitivity to TMZ.

Regulated by both growth-factor and cytokine levels, CK2 can directly regulate the activation of pathways associated with cell proliferation, migration, and angiogenesis, further contributing to GBM aggressiveness [31,40]. Zheng et al. reported that the pharmacologic inhibition of CK2 activity via the selective inhibitor, CX-4945, displayed an attenuated growth effect in an orthotopic model of GBM, warranting the therapeutic targeting of CK2 kinase activity [30]. Additionally, as CK2 is commonly amplified in GBM patients (33.7%), our study may support the potential use of CK2 levels as biomarkers of TTFields-treatment responsiveness [30]. We suggest that a decrease in CK2 activation measured by serine 129 AKT phosphorylation may serve as a biomarker of TTFields response.

We also showed in two models of GBM that the combination of PDGFR inhibition and TTFields treatment significantly increases cell death independently of TMZ sensitivity. Whether the benefit is additive or synergistic remains to be determined, as we only tested an estimated IC50 for the PDGFR inhibitor in our combinatorial studies. However, the results do suggest clinically relevant implications for the use of TTFields. The PDGFRA has been characterized as one of the most highly amplified gene in GBM tumors and serves as a negative prognostic factor for GBM-patient survival [41,42]. The molecular consequence of activated PDGFRA is exemplified through enhanced PI3K/AKT-, JAK/STAT-, and MAPK-pathway activity contributing to cancer-cell survival and overall disease severity [43,44,45]. The potential of TTFields to interact with such a crucial pro-survival receptor, tyrosine kinase, provides the basis on which to further consider combinatorial therapeutic strategies to overcome potential compensatory mechanisms. The stratification of patients based on PDGFRA amplification/mutational status may serve as a predictive biomarker for patients more inherently resistant to TTFields therapy. Additionally, as PDGFR has been shown to contribute to the pathogenesis of the proneural subtype of GBM, we hypothesize that tumors with elevated activity of this pathway may specifically benefit from TTFields and PDGFR inhibition [42]. The pro-survival role of PDGFR in the context of TTFields suggests that these patients may not respond to TTFields as a monotherapy, in which case the combined blockade of this pathway is necessary. However, the heterogeneity of GBM tumors further complicates this linear approach, necessitating continued investigation. We believe our findings may support the future consideration of clinical trials with TTFields and specific PDGFRA inhibitors.

Further exploration of the TTFields-induced changes in kinase activity that were distinct in our GBM models is warranted. For example, the kinases predicted to be differentially activated in TMZ-resistant cells included Akt1 in Jx22T and Src in U251T. The activation of the serine–threonine-kinase Akt pathway regulates survival, transcriptional- and protein-level regulation, and proliferation [46,47]. More specifically, Akt is activated in about 90% of all GBMs, necessitating therapies that can target this aberrantly activated pathway [41]. Additionally, Src kinases have also been negatively associated with GBM progression. A 2015 study by Lewis-Tufin et al. determined that Src-family kinases differentially promoted the growth and motility of GBM cells, and that the genetic modulation of these kinase family members reversed these affects [48]. Additionally, the Src kinases had varying predicted activity levels in radiation-therapy-selected GBM cells, which may mediate resistance phenotypes [25]. The determination of the effects of combining Akt or Src inhibitors with TTFields in different TMZ-resistant cell populations would be informative. The provision of further insights into how TTFields disproportionally target certain kinases that may hinder or aid cell growth responses is important. Additionally, to validate the clinical impact of kinases on TTFields sensitivity, an interesting avenue would be to investigate resected clinical specimens treated with TTFields to determine changes in the global kinome.

Beyond cell growth and survival, many other considerations of GBM-cell biology need to be considered as TTFields progress, such as the further exploration of the role that the physiologic and immune-cell microenvironments have on TTFields efficacy. The complexity of the tumor microenvironment has been shown to be disadvantageous in multiple therapies [49]. Physiologically, hypoxia and nutrient restriction mediate adaptive and cell-survival mechanisms that allow resistance to therapies [50,51]. Furthermore, as tumor-associated macrophages can aid in pro-tumorigenic response to therapies, more research is needed to determine the impact of TTFields on these specific cell populations [52]. Additionally, other therapy-resistant populations, such as radioresistant cells, also need to be considered as we continue to optimize TTFields for GBM treatment.

## 5. Conclusions

Overall, our study identifies novel mechanisms of kinase regulation by TTFields that can be exploited to enhance its therapeutic efficacy. As PDGFRα activity was predicted to be induced following TTFields, we present findings suggesting that PDGFR inhibition with crenolanib alongside TTFields administration provides significant combinatorial benefits. Additionally, our kinomic data present various other signaling cascades that are regulated by TTFields or may mediate sensitivity, warranting further validation and investigation. We believe that our findings may leverage the clinical utilization of TTFields, as they can be used to identify more precise combinatorial strategies or predictive biomarkers to stratify patient enrolment.

## Figures and Tables

**Figure 1 cells-12-02171-f001:**
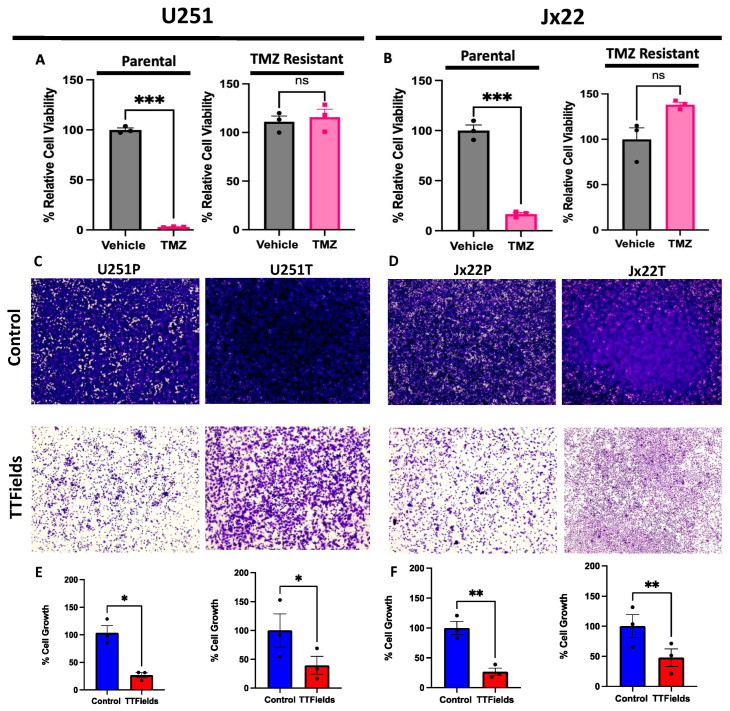
TTFields Decrease the Growth of Parental and TMZ-Resistant GBM Cells. Representative analyses of cell -line models (U251) and GBM-patient-derived xenografts (Jx22) confirmed for TMZ sensitivity or resistance following 7-day treatment with 150 µM TMZ ((**A**,**B**), respectively). Representative images of TMZ-sensitive and -resistant U251 cells (**C**) and Jx22P PDX (**D**) subjected to control culture conditions or 200 kHz of TTFields for 72 h, with corresponding quantifications for U251 (**E**) and Jx22 (**F**) indicated below respective images. TMZ-sensitivity analyses were performed in biological triplicates with representative graphs from one experiment shown and analyzed using an unpaired t-Test. TTFields-growth analyses were performed in biological triplicates (conducted with technical duplicates) and data were compiled and analyzed using a paired t-test. Data are displayed as means ± SEM. ns = not significant, * *p* ≤ 0.05, ** *p* ≤ 0.01, *** *p* ≤ 0.001. Magnification = 4×.

**Figure 2 cells-12-02171-f002:**
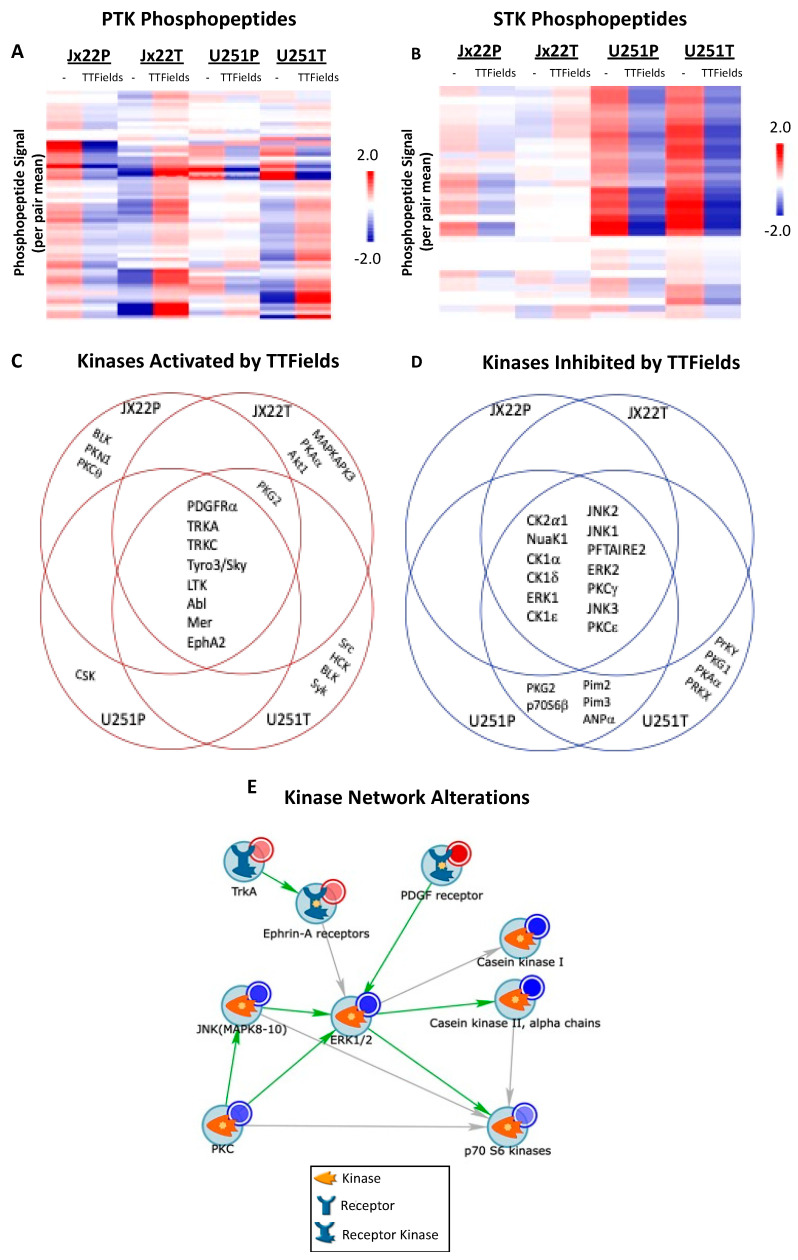
Phosphorylation Signatures are Altered Following Exposure to TTFields and Predict Differential Kinase Activity. Kinomics using harvested protein lysates from either control or 200-kilohertz-treated GBM cells was performed using PamChip arrays detecting specific peptide phosphorylation of tyrosine residues (**A**) or serine/threonine residues (**B**). Phosphopeptide signal of high-signal peptides are displayed as heatmaps (colored by Log2 Signal, change from per-pair mean). Predictions of both activated (**C**) and inhibited (**D**) kinase activity performed utilizing BioNavigator software (Pamgene, Den Bosch, Netherlands v6.3.67.0) and Kinexus databases with altered kinases (mean score > 1.0) are displayed as overlapping or independently altered [26,27,28,29]. The highest-scoring kinases (mean kinase score > 1.5) altered by TTF were network-modeled (**E**) using an AutoExpand algorithm with maximum nodes set at 50.

**Figure 3 cells-12-02171-f003:**
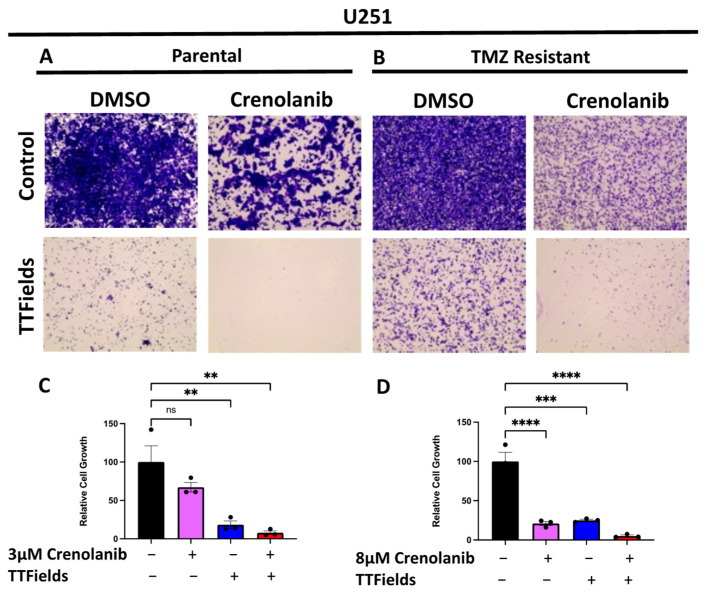
Treatment with TTFields and the PDGFR Inhibitor, Crenolanib, Displays Enhanced Benefit in Reducing Glioblastoma Cell Line Growth. TMZ-sensitive U251P (**A**) and TMZ-resistant (**B**) were treated for 72 h with either DMSO control, 3 µM or 8 µM crenolanib, 200 kHz of TTFields, or the combination of TTFields and crenolanib. Representative images and compiled data for U251P (**C**) and U251T (**D**) are displayed. Statistical analyses between groups were performed using one-way ANOVA with Tukey’s multiple comparison. Data are displayed as means ± SEM (*n* = 3 biological replicates performed in technical duplicates). ** *p* ≤ 0.01,*** *p* ≤ 0.001, **** *p* ≤ 0.0001. Image objective = 4×.

**Figure 4 cells-12-02171-f004:**
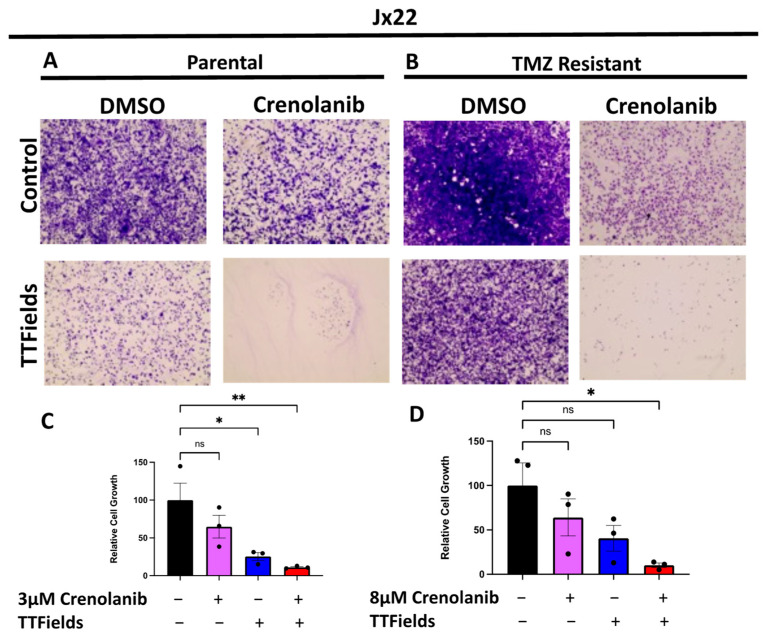
Reduction in Cell Growth is Observed in Patient-Derived Xenografts following Combinatorial Treatment with TTFields and Crenolanib. Jx22-patient-derived-xenograft cells were exposed to control or 200 kHz of TTFields in the presence or absence of crenolanib. Representative images from a single experiment from TMZ-sensitive cells (**A**) and TMZ-resistant cells (**B**) are displayed. Growth analyses were performed in biological triplicates (conducted with technical duplicates) and compiled data for the Jx22P (**C**) and Jx22T (**D**) were analyzed using a one-way ANOVA with Tukey’s multiple comparison. Data are displayed as means ± SEM. * *p* ≤ 0.05 ** *p* ≤ 0.01, image objective = 4×.

## Data Availability

Kinomic data will be made available upon request.

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
