# Peer review of "Tumor Treating Fields Alter the Kinomic Landscape in Glioblastoma Revealing Therapeutic Vulnerabilities"

_cells, 2023, doi:10.3390/cells12172171_

Round 1

Reviewer 1 Report

The authors address a highly important aspect in the management of GBM, as to what extent TTFields can also be effective in temozolomide – resistant cells. Clinically, this question is crucial since treatment of relapsing GBM showing reduced or no response to systemic treatment with alkylating agents is one of the biggest challenges. The authors provide an excellent study addressing the potential influence of TTFields on kinase–related signalling by performing kinomic profiling. The paper is very well written and deserves publication. There are only a few aspects that need to be addressed before being published in Cells.

1.      The wording of the abstract is somewhat overly careful. A phase III study has demonstrated that GBM patients treated with TTFields show significantly longer progressionfree- and overall survival. This should be phrased accordingly instead of “may be improved…etc.”

2.      The authors mention in the abstract that TTFields are currently approved for recurrent GBM as monotherapy. Since 2015 TTFields are also approved for the treatment of newly diagnosed GBM as an adjunct to the Stupp protocol. This needs to be mentioned as it is in the introduction.

3.      Is the addition of crenolanib to the TTFields additive or synergistic? Is there a chance to quantify the interaction between the two treatment effects?

4.      Given the important role of PDGFRa activation demonstrated in this study, do the authors suspect a specific reaction pattern in the proneural subtype of GBM, in which the PDGFR signaling pathway is altered?

5.      Do the authors plan to look into the tissue of recurrent GBM specimens resected following TTFields treatment to verify these results regarding the impact on the kinomic profile?

6.      In the conclusion, the authors state that their results could be used as predictive biomarkers. How exactly should that be integrated into the clinical routine? Do they propose to test for the activation of PDGFRa in the patients prior to TTFields treatment?  How should an elevation of kinases post TTFields treatment be assessed? By re–resection, which is done in only 20-25% of all GBM cases?  Alternatively, are there any liquid biopsy approaches available for this aspect (I am not aware of any such strategies).  

Author Response

Both reviewers indicated the need to better outline TTFields use in the clinical setting in both the abstract and introduction: Reviewer 1: “The wording of the abstract is somewhat overly careful…. This should be phrased accordingly instead of “may be improved…etc.”” Reviewer 1: “Since 2015 TTFields are also approved for the treatment of newly diagnosed GBM as an adjunct to the Stupp protocol. This needs to be mentioned as it is in the introduction.” Reviewer 2: “Authors should present more details, including the numerical survival benefit of TTFields in months.”

Response: We thank the reviewers for these suggestions and have modified the abstract to better detail the benefit of TTFields and to more accurately present its clinical utilization. We have also expanded on the numerical survival benefits of TTFields: “A recent meta-analysis investigating clinical outcomes utilizing TTFields reported that, based on 1309 cases spanning 14 studies, there was a significant benefit in one year survival rates for TTFields treated patients (>60%) as compared to untreated (<40%) warranting continued utilization[11]. Guzauskas et. al further provided an integrated epidemiological approach using TTFields EF-14 clinical trial data to predict survival probability of GBM patients[12]. Based on their analysis, it was predicted that patients alive 2-years post first starting TTFields have a 20.7% probability of surviving 10-years after diagnosis[12].”

Reviewer 1 suggested that we further characterize the benefits of crenolanib to TTField administration: “Is the addition of crenolanib to the TTFields additive or synergistic? Is there a chance to quantify the interaction between the two treatment effects?”

Response: We agree with the reviewer that this is very interesting area of investigation. However due to the current experimental setup for the TTFields system, we are limited to only a small number of groups per experiment.  This limits our capacity to do a thorough curve of treatment combinations to specifically define additive or synergistic relationships.  To acknowledge this limitation, we have added the following text to the discussion:  “Whether the benefit is additive or synergistic remains to be determined, as we only tested an estimated IC50 for the PDGFR inhibitor in our combinatorial studies.”

Reviewer 1 provided suggestions for future avenues for this project based on our presented findings and potential clinical implications that we have addressed in the discussion:  “…do the authors suspect a specific reaction pattern in the proneural subtype of GBM, in which the PDGFR signaling pathway is altered?”  “Do the authors plan to look into the tissue of recurrent GBM specimens resected following TTFields treatment to verify these results regarding the impact on the kinomic profile?“ “In the conclusion, the authors state that their results could be used as predictive biomarkers. How exactly should that be integrated into the clinical routine? Do they propose to test for the activation of PDGFRa in the patients prior to TTFields treatment?  How should an elevation of kinases post TTFields treatment be assessed?”

Response: We appreciate the insightful considerations for this project and have modified the discussion to present some of these suggestions. Specifically, we now state in the discussion: “Additionally, as PDGFR has been shown to contribute the pathogenesis of the proneural subtype of GBM, we hypothesize that tumors with elevated activity of this pathway may specifically benefit from TTFields and PDGFR inhibition [42]. The pro-survival role of PDGFR in the context of TTFields suggests that these patients may not respond to TTFields as a monotherapy, therefore necessitating combined blockade of this pathway. However, the heterogeneity of GBM tumors further complicates this linear approach necessitating continued investigation.” Later in the discussion, we also indicate that:  “Additionally, to validate the clinical impact that kinases have on TTFields sensitivity, an interesting avenue would be to investigate resected clinical specimens treated with TTFields to determine changes in the global kinome.”

Reviewer 2 Report

Authors of the study investigated glioblastoma cell models, whether the treatment with TTFields alter the kinomic landscape of GMB, and to identify potential additional biomarkers and/or therapeutic targets. The following issues were found.

  1. Line 66-74: Authors should formulate  only the specific study questions in introduction, and the summary of study results should be removed.
  2. Authors indicated in introduction that both newly diagnosed and recurrent GBMs can benefit from TTField treatment. Authors should present more details, including the numerical survival benefit of TTFields in months. E.g., A recent meta-analysis also investigated this question: DOI: 10.3390/cancers15030880.
  3. Lines 101, 117, 118, and 139: the units are either missing a space or the exponent of the power is badly formatted
  4. Line 163-165: Please provide a proper Statistical analysis section for the paper. The used statistical methods, their applicability testing, whether p-value correction was used, etc. must be complete, and not listed below figures only.
  5. Figure 1: Second 'D' should be changed to 'E'.
  6. Lines 195-196, 270 and 285: "Image objective = 4x." Did authors mean “Magnification = 4x”?
  7. Figure 2A and 2B: please indicate what the x-axis on the heatmaps represent.
  8. The quality of Figure 2 is low, please update the text with a higher quality version. In particular, Figure 2F cannot be read in the current version.
  9. Line 242-247: these statement needs to be referenced.
  10. Please indicated what are the black circles by the SEMs on Figures 3C and 3D.
  11. Line 326-327: Although the authors also wrote it in the conditional mode, perhaps it is still worth further refining the wording here. Reviewer suggests that this sentence should be rephrased as a hypothesis, as the current study could not tested this question in a clinical setting.
  12. In general, a thorough read is necessary, there are several typos, spaces are missing.
  13. A thorough English read is necessary.
  14. Line 370-371: Authors have supplementary data, that are not reported here.

Minor editing of English language required

Author Response

Both reviewers indicated the need to better outline TTFields use in the clinical setting in both the abstract and introduction: Reviewer 1: “The wording of the abstract is somewhat overly careful…. This should be phrased accordingly instead of “may be improved…etc.”” Reviewer 1: “Since 2015 TTFields are also approved for the treatment of newly diagnosed GBM as an adjunct to the Stupp protocol. This needs to be mentioned as it is in the introduction.” Reviewer 2: “Authors should present more details, including the numerical survival benefit of TTFields in months.”

Response: We thank the reviewers for these suggestions and have modified the abstract to better detail the benefit of TTFields and to more accurately present its clinical utilization. We have also expanded on the numerical survival benefits of TTFields: “A recent meta-analysis investigating clinical outcomes utilizing TTFields reported that, based on 1309 cases spanning 14 studies, there was a significant benefit in one year survival rates for TTFields treated patients (>60%) as compared to untreated (<40%) warranting continued utilization[11]. Guzauskas et. al further provided an integrated epidemiological approach using TTFields EF-14 clinical trial data to predict survival probability of GBM patients[12]. Based on their analysis, it was predicted that patients alive 2-years post first starting TTFields have a 20.7% probability of surviving 10-years after diagnosis[12].”

Reviewer 2 Line 66-74: Authors should formulate only the specific study questions in introduction, and the summary of study results should be removed.

Response:  We have removed the text as requested.

Reviewer 2 indicated several minor concerns about the text:  “Lines 101, 117, 118, and 139: the units are either missing a space or the exponent of the power is badly formatted.”  “Figure 1: Second 'D' should be changed to 'E'.” “Line 242-247: these statement needs to be referenced.” “Line 370-371: Authors have supplementary data, that are not reported here.”

Response:  We sincerely thank Reviewer 2 for their careful review and have corrected the errors that were noted.

Reviewer 2 asked us to clarify “what are the black circles by the SEMs on Figures 3C and 3D.”

Response:  We have checked Figure 3 on both a PC and Mac and cannot see any black circles other than the data points that are the 3 biological replicates.  We will confirm with the editors that there are no unusual images on this figure as we understand that conversion errors can occur. 

Reviewer 2 indicated that “the quality of Figure 2 is low, please update the text with a higher quality version. In particular, Figure 2F cannot be read in the current version.”  “Figure 2A and 2B: please indicate what the x-axis on the heatmaps represent.”  

Response: We have now provided a higher quality image for our kinomic data with a more clearly labeled x-axis.

Reviewer 2 indicated that we need to “..provide a proper Statistical analysis section for the paper. The used statistical methods, their applicability testing, whether p-value correction was used, etc. must be complete, and not listed below figures only.”

Response:  We have expanded our statistical analysis section with the following: “For basal growth analysis with individual treatment with TMZ or TTFields, a paired T-test was performed based on the following met assumptions: independence, continuous outcome variable, normal distribution (non-significant Shapiro-Wilks tests). For combined TTFields and crenolanib growth assessments, statistical analyses were performed using a one-way ANOVA with Tukey’s multiple comparison based on the following met assumptions: independence, normal distribution (non-significant Shapiro-Wilks tests), homogeneity of variance (non-significant Brown-Forsythe tests), and continuous outcome variable. Reported p-values are adjusted based on alpha = 0.05.”

Round 2

Reviewer 2 Report

Authors addressed all my concerns.